# Data-driven reaction coordinate discovery in overdamped and non-conservative systems: application to optical matter structural isomerization

Shiqi Chen [1,2], Curtis W. Peterson [1,2], John A. Parker [2,3], Stuart A. Rice [1,2], Andrew L. Ferguson [4✉] & Norbert F. Scherer [1,2✉]

Optical matter (OM) systems consist of (nano-)particle constituents in solution that can self-organize into ordered arrays that are bound by electrodynamic interactions. They also manifest non-conservative forces, and the motions of the nano-particles are overdamped; i.e., they exhibit diffusive trajectories. We propose a data-driven approach based on principal components analysis (PCA) to determine the collective modes of non-conservative over-damped systems, such as OM structures, and harmonic linear discriminant analysis (HLDA) of time trajectories to estimate the reaction coordinate for structural transitions. We demonstrate the approach via electrodynamics-Langevin dynamics simulations of six electrodynamically-bound nanoparticles in an incident laser beam. The reaction coordinate we discover is in excellent accord with a rigorous committor analysis, and the identified mechanism for structural isomerization is in very good agreement with the experimental observations. The PCA-HLDA approach to data-driven discovery of reaction coordinates can aid in understanding and eventually controlling non-conservative and overdamped systems including optical and active matter systems.

[1] Department of Chemistry, University of Chicago, Chicago, IL, USA. [2] James Franck Institute, University of Chicago, Chicago, IL, USA. [3] Department of Physics, University of Chicago, Chicago, IL, USA. [4] Pritzker School of Molecular Engineering, University of Chicago, Chicago, IL, USA. ✉email: andrewferguson@uchicago.edu; nfschere@uchicago.edu

A major goal of chemical research is the determination of the details of atomic rearrangements, bonding, and reaction pathways of molecules[1]. Contrary to the simplified "general chemistry" perspective, true molecular reaction pathways are multi-dimensional and challenging to represent and visualize. The committor probability for a two-state system – the statistical probability that a particular configuration will transition into the product basin before the reactant basin – is the optimal reaction coordinate in that it is perfectly correlated with, and indeed defines, the extent of reaction[2]. As a purely statistical measure, the committor does not provide any configurational or physical understanding of the reaction mechanism, however simplified, dimensionally reduced, configuration-based reaction coordinates are valuable in defining and quantifying the important collective motions driving transitions between metastable configurations[3]. A traditional approach to quantifying the important dynamical fluctuations of a system in a metastable configuration is to project them onto the leading (vibrational) normal modes of the system as a basis set that allows characterization of the soft collective fluctuations[4]. Alternatively, transition path sampling (TPS) and related methods[5,6] can determine reaction paths in high dimensional systems by identifying high-probability reactive paths.

In addition to the aforementioned methods for molecular systems, atomic and molecular Van der Waals clusters made in molecular beams, and colloidal clusters in solution represent classes of materials that form close-packed configurations that are held together by Van der Waals (i.e., dispersion) interactions[7], depletion forces[8–10], or Casimir type interactions[11]. Their configurations are well-determined by repulsive "hard-sphere" interactions as described by Weeks, Chandler, and Anderson (WCA) theory[12]. Since the interactions are typically short ranged, the transitions between different structural isomers are often single particle moves[9,11] or correlated few particle moves[13] that are often interpretable in terms of broken 'bonds' and involve a small number of degrees of freedom. Although the transition state configurations are a challenge to study in atomic clusters due to their small spatial scale and short lifetime, the size of colloidal systems readily allow conventional optical microscopy and fast imaging to be used for visualization of particle trajectories.

Optical matter (OM) is a type of material or molecule-like structure in which the constituents (e.g. nanoparticles (NPs) or micron-scale particles) are bound together by electrodynamic interactions[14,15]. A fundamental aspect of OM structures is that they tend to occupy inter-particle distances that are integer multiples of the wavelength of the incident optical field. This is particularly clear when the constituents are NPs that are smaller than the wavelength of light[15,16]. This interaction, known as optical binding[14,17,18], allows formation of regular OM configurations (e.g. 2D arrays with trigonal symmetry or anisotropic arrays with rectangular lattice configurations) with minimal optical information, generally only the overall shape, polarization, phase, and power of the optical beam[17] and the resulting particle configurations are readily visualized by optical microscopy[15,19–21]. These systems are of particular interest in optical physics due to their manifestation of non-reciprocal forces, collective (correlated) interactions, and many-body effects in their electrodynamic interactions[22–26] and can also serve as useful mesoscale analogs with which to study atomic-level chemical processes[27].

It is known that long range interactions manifest structural and dynamic correlations in driven dense colloidal solutions due to hydrodynamic interactions[28], and in quantum-dot perturbed molten salt solutions[29]. OM configurations self-organize in optical traps in solution and achieve new collective properties due to their electrodynamic interactions and long range (periodic) potentials[14,15,17,18]. The long range interactions create a new richness for understanding isomerizations of OM configurations

as they affect the energetics globally and imbue OM systems with many-body interactions and physics. Rational control of the configuration of optical matter requires solving a many-body problem by effective and practical methods.

Figure 1 demonstrates the similarities between molecular and optical matter systems. The probability distribution of a molecular configuration in the neighborhood of the potential minimum can be well approximated by a multivariate Gaussian distribution with a covariance matrix governed by the Hessian matrix $H$ at the minimum of the potential defined as:

$$H_{ij} = \frac{\partial U(\mathbf{r})}{\partial r_i \partial r_j} \tag{1}$$

where $\mathbf{r}$ represents the Cartesian coordinates of the molecular configuration and $U$ is the potential function. When the potential is harmonic, the normal modes of the system can be determined from the covariance matrix[4] (a detailed derivation is provided in Supplementary Discussion). Consider the asymmetric stretching mode of a water molecule shown in Fig. 1a–d as a simple example. The coordinate displacement exemplified in this mode is involved in the dissociation and autoionization reactions of water[30]. A second example is boat-to-chair isomerization of cyclohexane in which particular normal modes dominate others in effecting the isomerization transition[31]. Optical matter systems in optical traps in solution can undergo structural transitions. For example, Fig. 1e–g illustrate a transition in a 6-particle OM

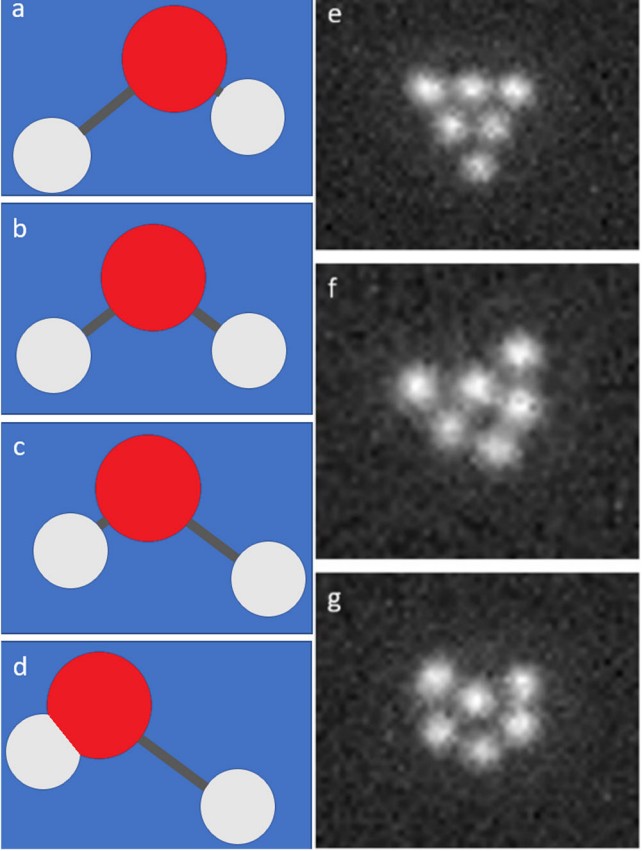

**Fig. 1 An analogy between molecular normal modes and the collective modes of optical matter systems. a–d** The antisymmetric stretching mode of the water ($H_2O$) molecule. **e–g** Experimentally measured "instantaneous" OM configurations associated with a transition between triangle (**e**) and chevron (**g**) via and intermediate higher energy transition configuration (**f**) in the 6-particle optical matter system that forms and fluctuates in a converging Gaussian optical beam that is circularly polarized.

system from a "triangle" to a "chevron" configuration through a transition state. (Experimental details for creating the OM system and optical microscopy visualization are given in Methods section) However, compared to molecular systems, besides friction, the external electrodynamic forces of the OM system in solution are non-conservative and overdamped[32]. Is it possible, then, to find an analog of normal mode analysis for OM systems in solution? In answer, we propose that this transition can be described by the collective modes of the OM system. The primary goal of this paper is to develop a collective coordinate analysis for non-conservative, overdamped systems, and demonstrate the approach in defining a reaction coordinate and transition mechanism in an OM system's structural isomerization.

Normal modes are orthogonal collective motions of particles that carry independent contributions to the system energy. The conventional definition of normal modes is valid only for harmonic particle–particle interactions. There is no formal mechanical definition of normal modes for overdamped and non-conservative systems. Consider the following Langevin equation:

$$m\frac{\mathrm{d}^2\mathbf{r}}{\mathrm{d}t^2} = \mathbf{F}_{\mathrm{ext}}(\mathbf{r}, t) - \xi\frac{\mathrm{d}\mathbf{r}}{\mathrm{d}t} + \boldsymbol{\eta} \qquad (2)$$

where $\mathbf{r}$ is the position, $m$ is the mass, $\mathbf{F}_{\mathrm{ext}}$ is the external force field, $\xi$ is the friction coefficient, and $\eta$ is the random force. Only when $\xi = 0$ and $\frac{\partial F_{\mathrm{ext},x}}{\partial y} = \frac{\partial F_{\mathrm{ext},y}}{\partial x}$ can normal modes be well-defined. It should be emphasized that the non-conservative nature of the OM system refers to the non-conservative (external) electrodynamic force field $\mathbf{F}_{\mathrm{ext}}$[32]. Renson and Kerschen have defined nonlinear normal modes in underdamped systems (where $\xi$ is not zero but not large enough to neglect the acceleration term)[33]. David and Jacobs have used principal component analysis (PCA) to study the large-scale fluctuations in molecular and colloidal systems[34]. Zaccone, et al. and co-workers have used instantaneous normal modes (INM) and the vibrational density of states in liquids to analyze none-affine dynamics of amorphous materials such as glassy polymers[35–38]. These two types of systems can be overdamped but conservative ($\xi$ is large enough to neglect the acceleration term; $\frac{\partial F_{\mathrm{ext},x}}{\partial y} = \frac{\partial F_{\mathrm{ext},y}}{\partial x}$). See Supplementary Information for further discussion. Chattoraj et al.[39] have analyzed the eigenvalues and eigenvectors of the J-matrix, the first derivative matrix of the external force field, and found oscillatory solutions of motion that are particularly useful for studies of underdamped non-conservative systems ($\xi$ is not large enough to neglect the acceleration term; $\frac{\partial F_{\mathrm{ext},x}}{\partial y} \neq \frac{\partial F_{\mathrm{ext},y}}{\partial x}$).

In overdamped cases, however, there exists no oscillatory solution while the non-orthogonal eigenvectors of the J-matrix lead to intrinsically coupled collective modes that are complicated to analyze. See the Supplementary Discussion for additional discussion. In our approach we define collective modes in overdamped and non-conservative systems ($\xi$ is large enough to neglect the acceleration term; $\frac{\partial F_{\mathrm{ext},x}}{\partial y} \neq \frac{\partial F_{\mathrm{ext},y}}{\partial x}$) by carrying out PCA on the configurational trajectories based on the deviations of the OM constituent particles from a reference configuration. PCA diagonalizes the covariance matrix of the OM particle coordinates to define a linear transformation into a basis of non-local collective modes (principal components, PCs) ordered by the degree of configurational variance they contain[40]. The leading PCs correspond to collective degrees of freedom with large variance that typically characterize large-scale global rearrangements of the system, whereas the trailing PCs correspond to small-variance fluctuations around particular metastable configurations. The leading PCs are therefore anticipated to serve as good descriptors for transitions between metastable system configurations. The

PCs may further be formally converted into reaction coordinates using harmonic linear discriminant analysis (HLDA)[4].

In this work, we demonstrate this PCA-HLDA approach described above in an application to trajectories of the triangle-to-chevron transition, like the measured result of Fig. 1e–g, using combined electrodynamics-Langevin dynamics simulations[23,41] of a 6-particle OM system. We note that the local fluctuation trajectory required for our PCA-HLDA approach must be adequately long with sufficiently fine time steps. Although this analysis could in principle be done by particle tracking analysis of experimental data[42], the transitions are rare and must be sampled at rates that are higher than can be readily obtained even in relatively high speed (100's fps) video measurements. We determine the contributions of each PCA collective mode to the transition, employ HLDA to formulate a reaction coordinate from these modes, validate the reaction coordinate using committor probability analysis, and use our results to define the transition state ensemble and reaction mechanism. This PCA-HLDA approach is analogous to those used to describe molecular reaction dynamics, but it is herein applied to an overdamped and non-conservative OM system[4].

## Results

We demonstrate our methodology on an OM system, which is an open system; an assembly of particles subject to a persistent flux of an external electromagnetic field that induces the inter-particle interactions. The OM system is also overdamped owing to the size and mass of the 150-nm diameter silver nanoparticles such that the acceleration term in the Langevin equation becomes negligible. The primary data are from electrodynamics-Langevin dynamics (EDLD) simulations of a 6-particle optical matter system (see Methods section), and the results are corroborated with experimental data. We first describe the observed collective motions of the 150-nm diameter silver nanoparticles in a focusing (converging) optical trapping beam of this non-conservative system. Then we apply PCA to the deviations from the triangle (reactant) configuration obtained from EDLD simulation trajectories to estimate the collective modes to serve as a basis to describe the transition to the chevron (product) configuration. Finally, we apply HLDA to the collective modes projected from a trajectory to obtain reaction coordinates to study the triangle-to-chevron configurational transition.

**Large-scale collective modes.** The OM system consists of six silver nanoparticles of 150 nm diameter confined to a 2D plane and optically trapped by a circularly polarized focused laser beam, which makes the interactions isotropic in 2D[23,25]. The NPs tend to form metastable configurations that, to a first approximation, maximize the number of optical binding electrodynamic interactions, analogous to chemical bonds, with an inter-particle spacing of approximately one optical wavelength[43]. The particles therefore tend to adopt configurations based on a hexagonal lattice. While the optical binding energies can be several (2-10) $k_BT$ units of thermal energy[15,42,43], the OM configurations undergo spontaneous configurational transitions between these metastable configurations driven by both thermal fluctuations and by non-conservative optical forces. The two most probable configurations for six particles under a circularly polarized laser beam are a "triangle" (Fig. 1e) and a "chevron" (Fig. 1g). Because energy is not well defined in the OM system, by saying a configuration is more stable we mean that it is more probable (more commonly observed in trajectories obtained from both simulation and experiment).

The EDLD simulations are relatively inexpensive to conduct, allowing us to obtain long trajectories of the OM system to obtain

good statistical sampling of configurations with high (~1 μs) time resolution. A representative video of the 6-particle system undergoing a structural transition is shown in Supplementary Movie 1. The representative instantaneous configuration and deviations of the positions of the NPs from the stable triangle configuration shown in Fig. 2a–c indicates correlated motion of the particles over the course of the triangle-to-chevron reconfiguration. In Fig. 2a–c we mark the positions of the NPs in the stable triangle configuration with red crosses and the instantaneous location of the particles (i.e., at a particular point in a trajectory) over the course of a representative transition from the triangle to the chevron configuration by black circles. Each configuration is rotated so that the sum of its squared deviations to the triangle reference configuration (red crosses) is minimized. Figure 2d–f shows shows histograms of the particle positions collected in structural epochs during which the simulated 150-nm diameter silver nanoparticles transition from the triangle (reactant) configuration (Fig. 2d) to the chevron (products) configuration (Fig. 2f) via a transition state (Fig. 2e). The simulations performed here generate long trajectories of the 6-particle OM system that undergo many transitions between triangle and chevron. This transition conserves a mirror axis of symmetry in the OM structures. Therefore, a primitive and intuitive reaction coordinate to characterize the transition is the distance between the two particles on this mirror axis labeled "2" and "6" in Fig. 2a. By binning frames in the trajectories into 20 nm increments of this distance we define 22 windows spanning the transition from the triangle to the chevron

configuration. The empirical probability densities computed from histograms over all frames in each window are presented for the first, middle, and last window in Fig. 2d–f, respectively. The tight probability distributions in these probability density functions suggest that there are well-defined collective motions among the NPs as they execute the triangle to chevron transition. However, the "smearing" of some of the sites at the transition state and for the chevron make it ambiguous whether the reaction really proceeds along a simple 1D path, which motivates the PCA-HLDA approach. Supplementary Figure 1 shows the experimental counterpart of Fig. 2d–f obtained from microscopy measurements of a 6-particle OM system undergoing the analogous structural transition.

Note that the initial perfect triangle structure corresponds to a trigonal lattice, which is taken as the reference structure for quantitative analysis of the particle fluctuations and deviations from the lattice sites. The distribution of squared deviations from the reference lattice shown in Fig. 3 also suggest the existence of collective modes in the OM system. Figure 3a shows the cumulative density function (CDF) of lattice fitting deviation as the sum of squares of particle position deviations from the ideal triangle lattice sites (i.e., the red crosses in Fig. 2a) accumulated for 2087 simulation data points of the (local) fluctuations of the OM system in the triangle configuration at 20 mW laser power, 14,982 at 40 mW, 24,168 at 60 mW, and 39,172 at 100 mW. The simulation conditions are calibrated to actual laboratory experiments; e.g. corresponding to an incident optical power of 100 mW of the trapping laser. The squares of particle position

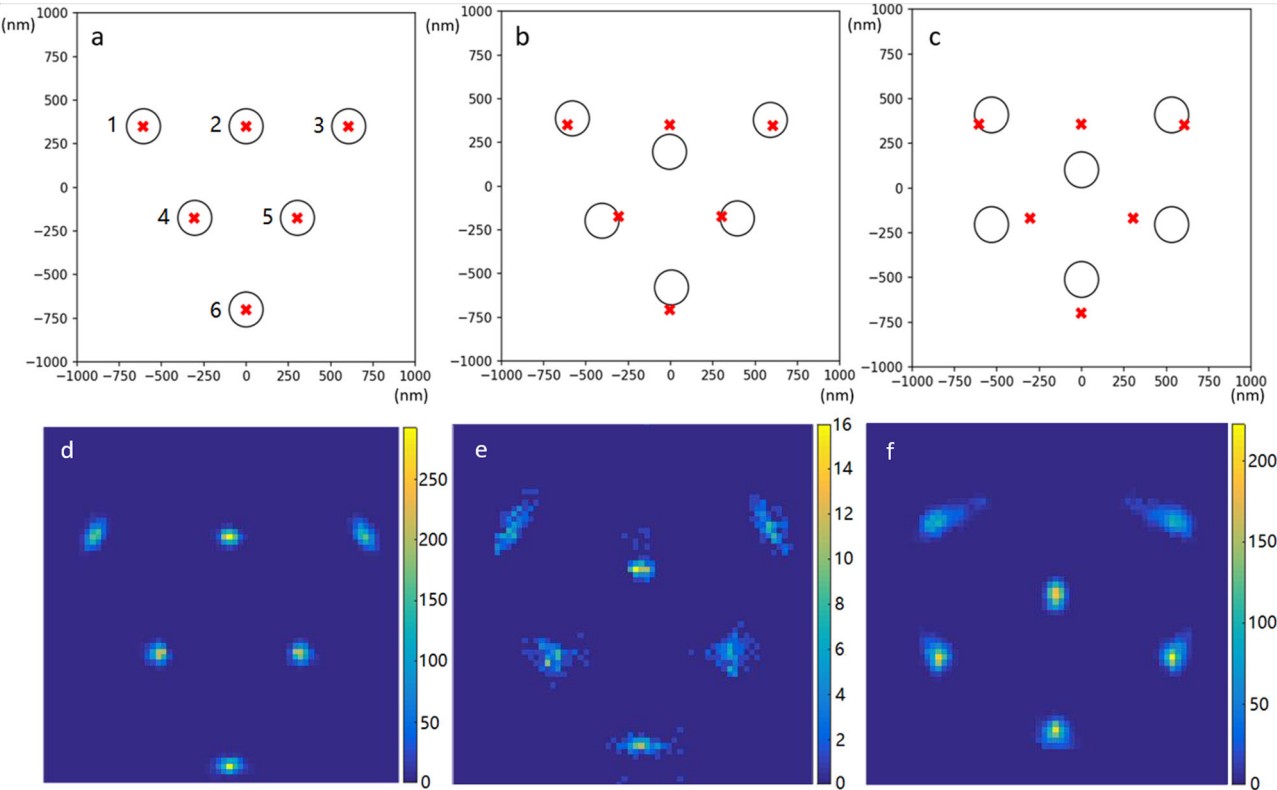

**Fig. 2 Particle positions and collective dynamics over the course of a triangle-to-chevron configurational isomerization. a–c** Mean deviations of particle position(s) (black circles) from the triangle configuration (red crosses) to the chevron configuration obtained over the course of a representative transition. **d–f** Empirical probability densities of particle positions compiled from 21,186 configurations at an optical power of 70 mW. The color scheme describes the number of configurations that contains a particle centered at a specific pixel. Configurations are binned into 22 windows of 20 nm in the distance between particles "2" and "6" (see panel **a**) and empirical probability density functions estimated by histogramming all configurations in each bin under a rotational and translational alignment to the mean particle positions (red crosses). The position probability density plots associated with **a–c** are presented in panels **d–f**, respectively.

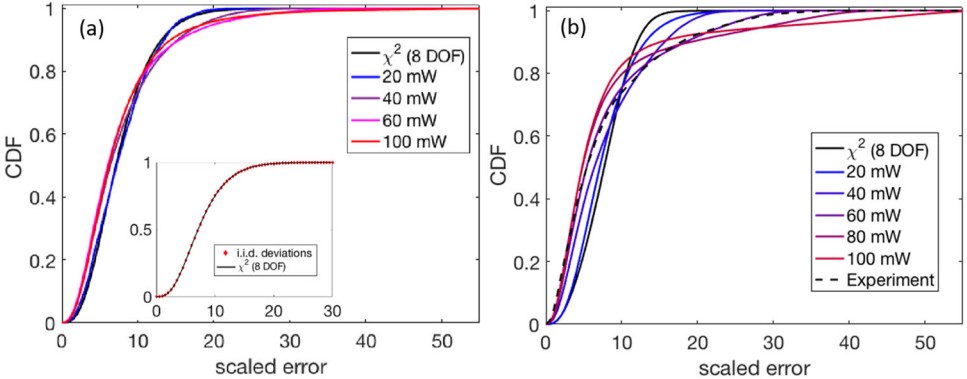

**Fig. 3 Cumulative density function (CDF) representation of distributions of lattice fitting displacement of the 6-particle system.** Cumulative density function (CDF) representation of distributions of lattice fitting of the 6-particle system in the local vicinity of the triangle configuration (**a**) and over the complete configuration space (**b**). The lattice fitting displacement is the sum of squares of particle position deviations of a configuration from the stable triangle lattice sites minimized over all possible translations, rotations, and the lattice parameter. **a** The fitting displacement computed from EDLD simulation trajectories of local fluctuations in the vicinity of the triangle configuration deviates from the 8 degree of freedom $\chi^2$ distribution that would be expected for particles executing uncorrelated i.i.d. Gaussian fluctuations (black curve) indicating the presence of collective structural modes. The inset shows a control simulation in which i.i.d. Gaussian fluctuations are imposed upon the particles (red diamonds) exactly follows the 8 DOF $\chi^2$ distribution. The magnitude of the deviation of the CDF fitting displacement distribution from the 8 DOF $\chi^2$ distribution (solid black curve) increases with optical trapping power in simulations conducted over the range 20–100 mW, indicating that the collective motions become more significant (and increase in magnitude) at higher optical powers. The corresponding probability density functions are shown in Supplementary Fig. 2. **b** The fitting displacement computed from EDLD simulation trajectories over the whole configuration space (i.e., not just local fluctuations) of the 6-particle OM system deviates from the 8 degree of freedom $\chi^2$ distribution. The magnitude of the deviation of the CDF fitting displacement distribution from the 8 DOF $\chi^2$ distribution (solid black curve) increases with optical trapping power in simulations conducted over the range 20–100 mW (solid colored curves) and experimental data gathered at an optical power of 50 mW (dashed black curve), indicating that the collective motions become more significant (and increase in magnitude) at higher optical powers.

deviations were calculated after minimizing all possible translations, rotations, and lattice parameters such that four of the 12 degrees of freedom are eliminated during the lattice fitting. (There are a total of $6 \times 2 = 12$ degrees of freedom for the 6-particle system constrained to a 2D plane.) If the particle motions of a system are uncorrelated and described by independent identical Gaussian distributions (i.i.d. Gaussian), then the CDF of squared translationally and rotationally minimized deviations should follow a $\chi^2$ distribution with $(12 - 4) = 8$ degrees of freedom (black curve). Calculations in which we impose i.i.d. Gaussian fluctuations upon the six particles (red diamonds, inset) do indeed follow exactly this trend. On the other hand, the dynamics and CDF obtained from EDLD simulation (colored lines, main figure) exhibit significant deviations from this trend with increasing beam power. This result suggests the inference that the particle fluctuations around the metastable triangle configuration are not independent but are coupled into one or more collective mechanical modes by the electrodynamic interactions induced by the incident laser and coherent field.

In Fig. 3b, we compute the deviation of the fitting displacement over 55,000 simulation data points harvested from the complete configurational space explored by the EDLD simulations (i.e., not just restricted to the local vicinity of the triangle configuration). Again, the fitting displacement distribution shows increasing deviations from the 8 degree of freedom $\chi^2$ distribution as the intensity of the optical field increases. This further supports the result that the extent of correlated motion of the particles increases with the intensity of the optical trapping beam. We note that deviations about the lattice sites are, of course, partially due to random uncorrelated fluctuations and partially due to correlated motions along collective coordinates and as the power increases the former become dominated by the latter.

**Principal component analysis and definition of collective modes.** Having confirmed the existence of collective motions in

the OM systems, we wish to study them quantitatively. We perform PCA on a single simulation trajectory that kept the lattice fitting displacement from the ideal triangle lattice to <250 nm to quantify the local collective fluctuations of the OM system around the triangle (reactant) configuration. Before PCA is carried out, each configuration is rotated according to the center of the field (i.e., the focused laser beam in the experiment) so that its fitting displacement with respect to the ideal triangular lattice is minimized. If this is not done the leading PCs will be contaminated by trivial rotations. The data set is then centered to a common origin by subtracting from the particle positions, the location of the center of the stable triangle configuration. The reason why other degrees of freedom such as translations are preserved is that the symmetry of the field is such that it is only invariant with rotation. However, with respect to the analogy with molecular systems that discard translations, even if unimportant degrees of freedom are taken into account, the result will not be harmed. Therefore, to be safe, only rotation is eliminated in data preprocessing.

The PCA identifies a linear transformation of the 12 degrees of freedom of the 6-particle system constrained to a 2D plane into a new basis of 12 orthogonal PCs arranged in order of decreasing configurational variance. The 12 PCs illustrated in Fig. 4 are ranked in order of largest to the smallest eigenvalues. PCs 1, 2, and 12 correspond to rigid body transformations: PC 1 and 2 correspond to global translations in directions indicated by the black arrows and PC 12 to global rotation. The remaining PCs correspond to the collective fluctuations around the (metastable) triangle configuration. Some modes, such as PC 3, are nearly symmetric, while others, such as PC 4, are nearly antisymmetric. The symmetry axes of symmetric modes are indicated by dashed lines.

We verified that the discovered PCs represent stable collective modes by showing that they are preserved upon analyzing long simulation trajectories. Figure 5a presents a heatmap reporting the inner product norms between the PCs computed from a 2-million-frame data set from a shorter trajectory and those

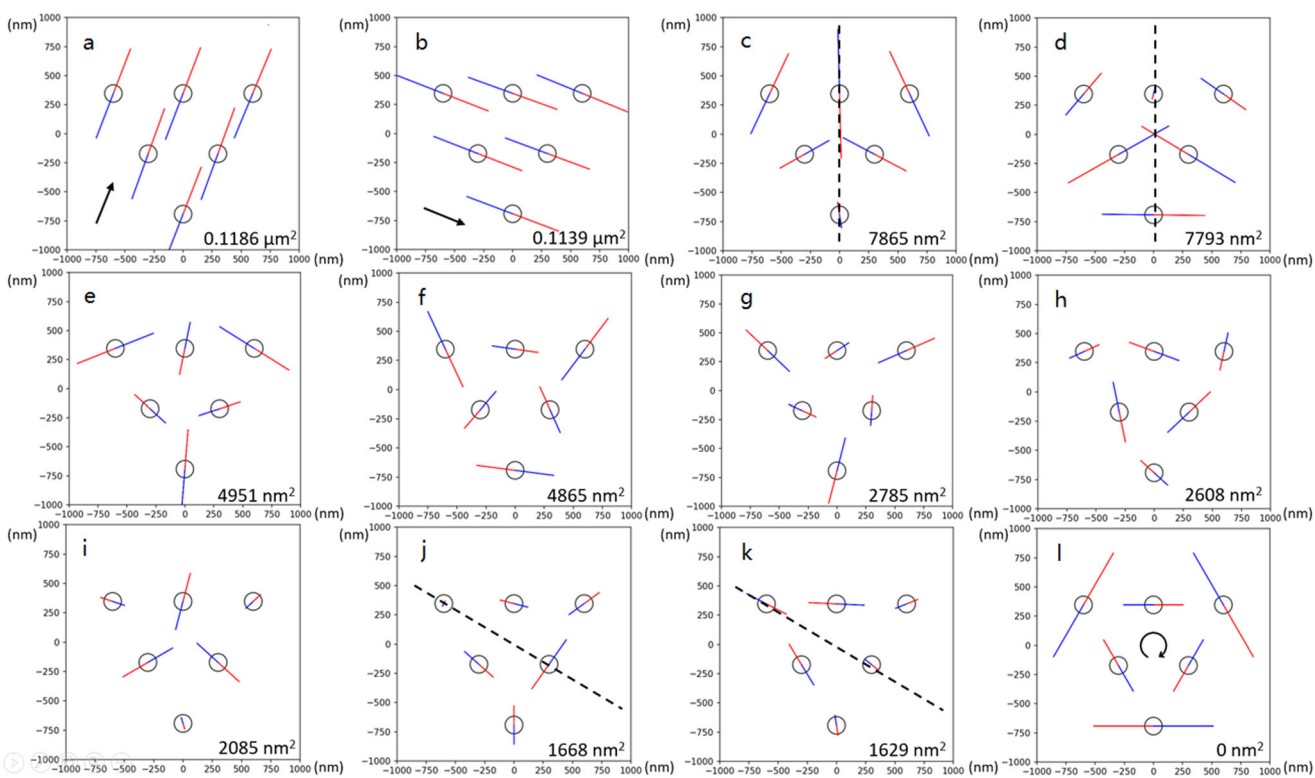

**Fig. 4 Principal components of the 6-particle triangle configuration.** Panels **a**–**l** correspond to PC 1 to 12, respectively. The colored solid lines depict the directions and magnitudes of the collective particle motions, and the color defines the sense (i.e. phase) of the motion; i.e., particles simultaneously move in the indicated directions for the same color. PCs 1 and 2 correspond to rigid translation and PC 12 to rigid rotation. For the non-rigid transformations, PCs 3–11, the length of the solid lines is proportional to the PCA eigenvalues, $\lambda_i$, which are reported in the bottom right of each panel. The value of the 12th eigenvalue is exactly zero due to elimination of rotational motion in application of PCA. Dashed black lines represent axes of symmetry. Black arrows indicate global translational or rotational motions.

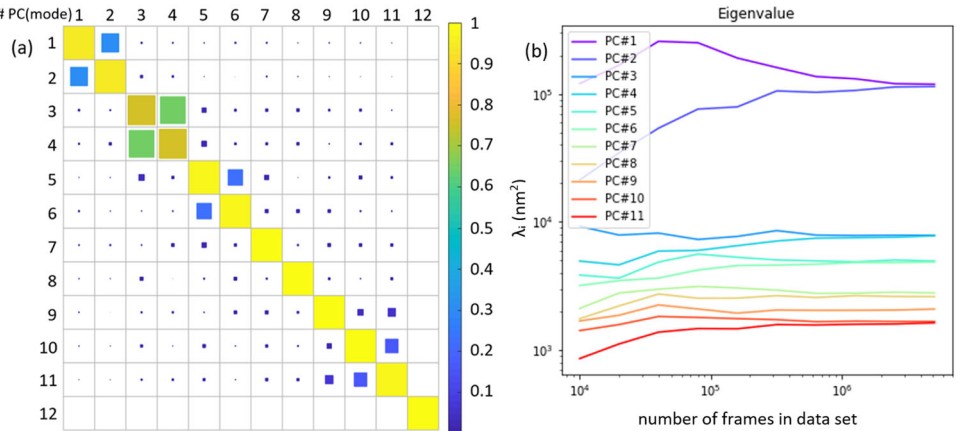

**Fig. 5 Convergence test of the PCs. a** The norm of the inner product between right singular vectors from PCA performed on a 2-million-frame (rows) and a 5-million-frame (columns) trajectory. **b** The eigenvalues $\lambda_i$ of the PCs $i = 1$–11 plotted against the number of frames in the analyzed data set.

computed from a 5-million-frame data set from a longer simulation. The matrix element (i,j) represents the norm of the inner product between the $i^{th}$ PC of the 2-million-frame trajectory and the $j^{th}$ PC of the 5-million-frame trajectory. If the PCs computed over the two data sets are identical, we would observe the identity matrix with elements (i,j) = 1 for i = j, and 0 otherwise. Therefore, the closer to the identity matrix Fig. 5a is the better converged we may assess the modes to be and therefore stable with increasing trajectory length. The matrix in Fig. 5a is close to identity with the exception of large off-diagonal elements between four pairs of PCs: (1,2), (3,4), (5,6), and (10,11). These

deviations from the identity matrix arise from the degenerate character of these four pairs (see Fig. 4) so that the corresponding PCs are resolved only up to an arbitrary angle within the eigenspace. The off-diagonal couplings we observe in Fig. 5a are due to linear mixing within this degenerate subspace. As such, the four degenerate pairs are robust between the 2-million-frame and 5-million-frame data sets, although the individual modes within these pairs are not, due to the arbitrary breaking of the degeneracy induced by PCA.

Figure 5b shows how the eigenvalues of the PCA converge as the size of the data set increases from 10,000 to 5 million frames.

We observe convergence of all eigenvalues $\{\lambda_i\}$ to stable values by around 2 million frames, indicating that our 2 million frame data set is sufficiently large to obtain converged and stable PCs. We also observe that the eigenvalues of the four degenerate PC pairs (1,2), (3,4), (5,6), and (10,11) converge to identical values within 4% error. Therefore, in light of the results in Fig. 5, we assert that the convergence test is successful, and that we have demonstrated that the PCs are indeed well-defined collective modes of the system.

**HLDA definition of reaction coordinates.** Having validated the PCs as proper collective modes, we seek to convert them into reaction coordinates for the triangle to chevron configurational isomerization using harmonic linear discriminant analysis (HLDA)[4]. The reaction coordinate is valuable in illuminating the transition mechanism, identifying the transition state ensemble, and providing a physically motivated measure of reaction progress that is vastly cheaper to compute than the committor probability and more configurationally informative.

The reaction coordinate generated by HLDA, for the transition from the triangle configuration to the chevron configuration, is defined as[4],

$$s_{\text{HLDA}}(\mathbf{R}) = (\mu_A - \mu_B)^T (\Sigma_A^{-1} + \Sigma_B^{-1}) d(\mathbf{R}) \qquad (3)$$

where $d$ is the 12-by-12 linear transformation matrix that converts particle positions $\mathbf{R}$ into the collective mode basis defined by the collective modes, and $\mu$ and $\Sigma$ are the mean and the covariance matrices of the collective modes. The subscripts $A$ and $B$ represent the triangle (reactant) and chevron (product) configurations, respectively. This expression can be considered as the projection of the coordinate in the collective mode basis onto the vector $\mathbf{W}^*$ that maximizes the Rayleigh ratio:

$$\mathbf{W}^* = \arg\max_{\mathbf{W}} \frac{\mathbf{W}^T \mathbf{S}_b \mathbf{W}}{\mathbf{W}^T \mathbf{S}_w \mathbf{W}} = (\Sigma_A^{-1} + \Sigma_B^{-1})(\mu_A - \mu_B) \qquad (4)$$

where $\mathbf{S}_b = (\mu_A - \mu_B)(\mu_A - \mu_B)^T$ is the between class scatter matrix, and $\mathbf{S}_w = (\Sigma_A^{-1} + \Sigma_B^{-1})^{-1}$ is the within class scatter matrix[4]. The vector $\mathbf{W}^*$ can therefore be interpreted as the direction along which the two classes are best separated[4].

Figure 6a shows the HLDA reaction coordinate, $s_{\text{HLDA}}(\mathbf{R})$, and the square of the projection of position deviation vector (projected variance) onto each of the collective modes for one

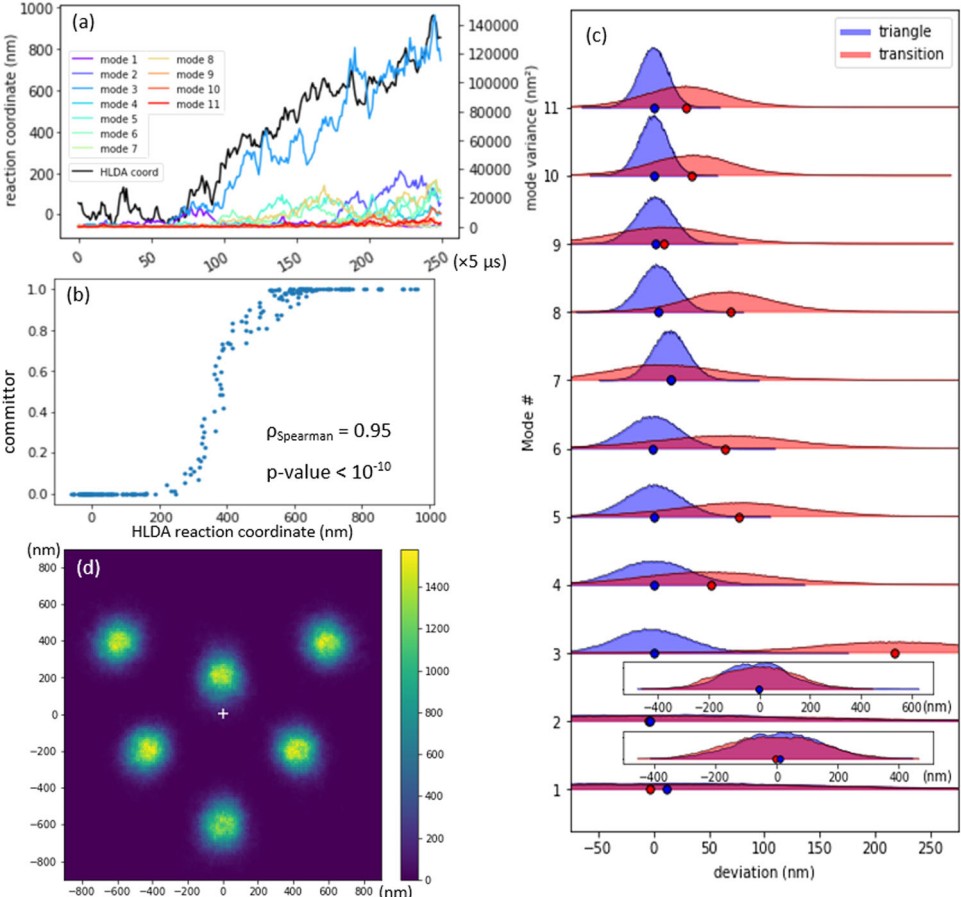

**Fig. 6 Validation and analysis of the PCA-HLDA reaction coordinate. a** Comparison of the HLDA reaction coordinate $s_{\text{HLDA}}$ and the variance of particle deviations along each of the 12 PCs over the course of a single triangle to chevron isomerization transition. **b** Calibration plot of the committor probability against the HLDA reaction coordinate for 250 selected configurations. The high rank-order correlation ($\rho_{\text{spearman}} = 0.95$) and low scatter validates the PCA-HLDA reaction coordinate as a good measure of reaction progress. **c** Distribution of particle deviations from the metastable triangle configuration projected onto all 11 non-trivial (omitting rigid rotation) PCs over the course of the transition for configurations with committor values 0.1–0.9 (red) and in the metastable triangle configuration with committor values <0.1 (blue). Means of each distribution are indicated by colored dots. PC 3 changes most markedly in moving from the metastable reactant basin of the triangle configuration to executing the isomerization transition. **d** Probability distribution of aligned particle positions computed over 395,860 configurations in the transition region with 300 nm < $s_{\text{HLDA}}(\mathbf{R})$ < 500 nm (committor probabilities in the range 0.1–0.9); the white cross indicates the average center of mass of these configurations. The color scheme describes the number of configurations that contains a particle centered at a specific pixel.

specific transition trajectory. Analogous plots for other representative trajectories and transitions are presented in Supplementary Fig. 3. The presented trajectory, which is typical of the majority of the transitions observed in simulations of the triangle-to-chevron transition, reveals a strong correlation between the HLDA coordinate (black curve) and PC 3 (blue curve) with Spearman correlation coefficient 0.93 and $p$-value $< 10^{-10}$, implying that PC 3 is the dominant collective mode that contributes to the HLDA reaction coordinate characterizing the triangle to chevron isomerization. This is intuitively reasonable considering the motions embodied in PC 3 (see Figs. 4 and 2d–f).

The committor probability associated with a particular configuration is calculated by computing the probability that a trajectory initialized from that specific configuration arrives at the chevron configuration (product) before arriving at the triangle configuration (reactant)[2,44]. A committor of zero indicates that the configuration is bound to arrive in the reactant basin before the product, a committor of unity means that it will first arrive in the product basin, and a committor of 0.5 that it has equal chance of first arriving in the reactant or product basin and is, by definition, a member of the transition state ensemble (TSE). We estimated the value of the committor for each OM structural configuration by initializing 420 trajectories at each of 250 configurations in a triangle-to-chevron transition trajectory, spanning the range of the HLDA reaction coordinate and computing the fraction that arrive in the chevron before the triangle. Figure 6b shows that the committor agrees well with the HLDA reaction coordinate because the committor changes monotonically with the HLDA reaction coordinate and the scatter in the plot has small variance. We identify from the plot, for example, that configurations with a $s_{\text{HLDA}}(\mathbf{R}) < 300$ nm in the vicinity of reactant, $s_{\text{HLDA}}(\mathbf{R}) > 500$ nm in the vicinity of product, and $300$ nm $< s_{\text{HLDA}}(\mathbf{R}) < 500$ nm as in the transition region. This committor analysis validates the HLDA reaction coordinate as a reliable structural measure of reaction progress, a useful means to identify the TSE, and as a tool to understand the isomerization mechanism.

We use the committor probability to perform two additional analyses of the configurational mechanism of the isomerization transition. First, to further quantify and illuminate the significance of PC 3 within the HLDA reaction coordinate, we extracted 1000 transition trajectories from the EDLD simulation trajectories and extracted configurations with committor probabilities in the range 0.1–0.9. Since the PCs form a complete basis set of the space of position deviation vectors, the position deviation vector of each configuration can be written as a linear combination of the PCs, in which the coefficients can be obtained by the following orthogonal projection:

$$\mathbf{r} - \mathbf{r}_0 = \sum[\mathbf{v}_i^{\text{T}}(\mathbf{r} - \mathbf{r}_0)]\mathbf{v}_i \qquad (5)$$

where, the $\mathbf{v}_i$'s are the principal components obtained from PCA, $\mathbf{r}_0$ is the coordinate of the triangle configuration, and $\mathbf{r}$ is the coordinate of a specific configuration in the trajectory. We then project the deviation vector of these transition configurations onto the 11 non-trivial collective modes – omitting the trivial rotation mode that was eliminated in our PCA analysis – to identify the distribution of configurational deviations from the triangle pattern in each of these modes over the course of the transition (Fig. 6c, red shading). We compare these distributions in each collective mode to those harvested from local fluctuation around the metastable triangle (Fig. 6c, blue shading). We plot the mean values of each distribution as a dot and present the precision of the mean values in Supplementary Fig. 4. This analysis clearly illustrates that PC 3 dominates the linear combination coefficients when configurational deviations are projected onto the PCs. In other words, configurational deviations in PC 3 change most markedly in moving from the

metastable triangle configuration to executing the isomerization transition relative to the other PCs. Note that the variance shown in Fig. 4 and the projection magnitudes shown in Fig. 6c are descriptions for two different processes. Figure 4 describes a trajectory that contains only the triangle configuration and its local fluctuations with no transition to another structure. Figure 6c describes an ensemble of trajectories of a specific transition from triangle to chevron. Therefore, the large variance of PC 4 described in Fig. 4 (local fluctuation of triangle) has no direct relation to whether it leads to the transition considered in Fig. 6c (transition from triangle to chevron). In addition, Fig. 6c only describes one specific transition starting from triangle (to chevron), so it is possible that mode 4 may dominate transitions to other states. Second, we collect the 395,860 configurations in the transition region with $300$ nm $< s_{\text{HLDA}}(\mathbf{R}) < 500$ nm and present the probability density of aligned particle positions in Fig. 6d. The transition state is identifiable as a configuration intermediate to the triangle and chevron configurations accessed by a collective motion along PC 3 as the dominant contributor to the HLDA reaction coordinate (cf. Fig. 4, red arrows).

## Discussion

Normal mode analysis is often used to study the statistics of the system configurations for conservative and undamped systems. However, this approach cannot be used for systems that are non-conservative, overdamped, and in which the configurational transformation involves large particle displacements. In this paper, we report an approach based on PCA to identify important collective fluctuations in non-conservative and overdamped systems and then use HLDA to transform these into a reaction coordinate for a configurational transition. We show that the PCs are stable collective modes and provide an interpretable basis for constructing and understanding the reaction coordinate. We demonstrate our approach in the triangle to chevron transition in numerical simulations of a 6-particle optical matter system and show that our results are consistent with experimental observations of the system. The HLDA reaction coordinate is shown to be valuable in resolving the transition mechanism and is validated as a robust reaction coordinate by committor analysis.

Data-driven discovery of reaction coordinates and kinetic transition rates is the first step in defining a kinetic network of the OM dynamics characterizing the metastable states and inter-state transition rates for the system. A number of methods and tools have been developed to understand the effect of incident field on configurations, stability, and non-conservative dynamics of OM arrays[45–47], including the spin angular momentum of light in optical tweezers used to introduce the driven spin of individual NPs[48–52], and inter-particle electrodynamic interactions that create orbital rotation in OM arrays[23,24,53]. Therefore, determining the kinetic network and quantifying the effect of these interventions upon OM kinetic networks offers a new route to engineer and control the stability and transitions of particular OM structural isomers with applications in optical matter machines[25].

In future work, we anticipate that the PCA-HLDA approach to reaction coordinate identification can be used to understand and quantify other configurational transitions in diverse optical matter systems and to construct kinetic networks for the global system dynamics. We anticipate that PCA-HLDA will prove particularly valuable for systems containing large numbers of particles where human intuition can often fail. There are several challenges associated with broader implementation of the PCA-HLDA method. The first is the magnitude of the data required. We show in Fig. 5b that the eigenvalues of the PCs take ~2 million frames to converge for this system (with a convergence resolution of 4%). While it is technically possible to acquire this

number of experimental frames (i.e. images of individual configurations) with the correct number of particles, a highly automated acquisition and analysis process would be required. Also, since our current implementation of PCA-HLDA was developed for short simulated trajectories where each particle is assigned to a specific lattice site while the experimental data will contain several rearrangements where lattice assignment will switch, developing a method to consistently assign experimentally obtained particle positions to specific lattice sites is not trivial. Therefore, we also anticipate that PCA-HLDA may, at least in the case of slow transitions that can be adequately characterized, be applied directly to experimental data sets by developing a method to consistently assign experimentally obtained particle positions to specific lattice sites for the trajectories obtained from experiments that involve rather frequent particle rearrangements.

We also anticipate a number of elaborations and improvements of PCA-HLDA. First, the method requires definition of a reference configuration to which the trajectory snapshots are aligned prior to application of PCA. In this work, we adopt the triangle configuration that is on hexagonal lattice sites as this reference is defined by a local minimum in the non-conservative force field. As the reactant configuration for the isomerization transition, this represents a natural choice, but we could also have adopted the product chevron configuration for this purpose. Determination of reference configurations that do not lie on well-defined lattice sites or exhibit high variance around a marginally metastable mean may be challenging, so that the PCA-HLDA approach may not be directly applicable for these configurations. Second, we see profitable integrations of PCA-HLDA with unsupervised nonlinear dimensionality reduction and clustering techniques, to first learn the metastable configurations and define which pairs are connected by configurational transitions in a data-driven manner and then use PCA-HLDA to identify reaction coordinates for transitions between each reactive pair.

Transitions involving indirect pathways, as shown in Supplementary Fig. 5, are also observed for the triangle-to-chevron isomerization. However, they are infrequent and are also not included in the HLDA analysis presented. Such pathways with intermediates will be considered in detail in future work. Apart from optical matter systems, the approaches presented here are promising in other non-conservative and overdamped systems such as active matter systems. Compared to optical matter, active matter is driven out of thermal equilibrium by stored or locally supplied free energy[26]. Approaches by Speck have used the work required to deform a certain volume of active matter to derive the dynamics of the active matter system[54], and work by Takatori and Brady has focused on an effective free energy for active particles[55]. With our research, it can be expected that the dynamics of active matter can be further explored.

## Methods

**Simulations**. Electrodynamics-Langevin dynamics (EDLD) simulations were performed with the Generalized Multiparticle Mie Theory (GMMT) using the MiePy software developed by the Scherer Lab[25,56]. Silver NPs with 150 nm diameter were used as the material constituents of the optical matter configurations. The nanoparticles were illuminated with a defocused, converging right-hand circularly (RHC) polarized Gaussian beam with a width $w = 2500$ nm, power $P = 50$ mW (except Fig. 3, where $P$ is varied), and defocus equal to the Rayleigh range, $z = 0.5kw^2$, where $k = 2\pi n_b/\lambda$ and $n_b$ is the index of refraction. These field/beam conditions allowed formation of stable 6-nanoparticle optical matter (OM) arrays even in the presence of thermal noise/forces. The electrodynamic forces were passed into an overdamped Langevin equation to integrate the equation of motion for the OM array with a 5-µs time-step using a simple first-order Euler integrator[57]. Two hours are required for a one-million-time-step trajectory on a 2.4 GHz Intel E5-2680 v4 CPU.

**Experiments**. Experiments were conducted using a single-beam circularly polarized optical tweezer in an inverted microscope setup as described in[15,23]. A dilute water solution with a mixture of polyvinyl pyrrolidone (PVP)-coated 150-nm Silver

NPs was used. A continuous wave (CW) Ti-Sapphire laser beam ($\lambda = 800$ nm) was focused near the glass cover-slip using a 60x microscope objective, pushing a small number of NPs toward the glass surface. The laser power was 200 mW before entering the microscope, where additional power is lost before focusing towards the sample. A spatial light modulator (SLM) was used to slightly defocus the trapping beam such that it was converging at the sample. Electrostatic repulsion between the ligands on the NPs and the glass cover-slip balances the radiation pressure, resulting in a 2D trapping envrionment.

**Lattice fitting**. Given a certain optical matter configuration with the number of particles, N, we want to find the best set of N sites on a 2d hexagonal lattice pattern that is closest to the given configuration and the positions of the particles in the given configuration be $\mathbf{r}_1, \mathbf{r}_2, ..., \mathbf{r}_N \in \mathbb{R}^2$. Let the positions of the hexagonal lattice sites be $\mathbf{s}_1, \mathbf{s}_2, ..., \mathbf{s}_N \in \mathbb{R}^2$. The formula of minimization of the fitting error is

$$Err^* = \left[ \min_{a \in \mathbb{R}^+, \mathbf{r}_0 \in \mathbb{R}^2, \hat{\mathbf{R}} \in SO(2), \pi \in \Pi} \sum_{j=1}^{N} \left| \hat{\mathbf{R}}(\mathbf{r}_j + \mathbf{r}_0) - a\mathbf{s}_{\pi(j)} \right|^2 \right]^{\frac{1}{2}} \quad (6)$$

where $\Pi$ is the set that contains all injections from 1, 2, ... N to $\mathbb{Z}^+$. Here, $a$ is the lattice parameter, $r_0$ is the translation vector, $\hat{\mathbf{R}}$ is the rotation matrix, and $\pi$ is the lattice assignment. Considering the symmetry of the lattice sites, the parameter domain can be further limited by forcing $\pi(1) = 1$ and $\mathbf{r}_0$ within the primitive cell. Then, the parameter space is discretized and optimized to get the best assignment between the particles and the lattice sites, $\pi^*$.

Next, with the assignment in hand, an analytical solution of the optimal translation, rotation, and lattice parameter can be found. Let $p_j = \mathbf{r}_j[1] + \mathbf{r}_j[2]i$ and $q_j = \mathbf{s}_{\pi(j)}[1] + \mathbf{s}_{\pi(j)}[2]i (j \in \{1, 2, ..., N\})$, so $\mathbf{p}, \mathbf{q} \in \mathbb{C}^N$. The fitting error expression can be transformed to the one below.

$$Err^* = \left[ \min_{a \in \mathbb{R}^+, p_0 \in \mathbb{C}, \theta \in [0, 2\pi)} \left| e^{i\theta}(\mathbf{p} + p_0\mathbf{1}) - a\mathbf{q} \right|^2 \right]^{\frac{1}{2}} \quad (7)$$

Where $p_0$ is the translation, $\theta$ is the rotation angle, $\mathbf{1}$ is the vector of all ones, and $a$ is the lattice parameter. The superscript "*" indicates the optimal value. The detailed derivation of the results below is presented in Supplementary Methods. If translation is exempted from optimization, then $p_0^* = 0$. Otherwise:

$$p_0^* = \frac{\mathbf{1}^T}{N}(\mathbf{q} - \mathbf{p}) \quad (8)$$

Let $\mathbf{p}' = \mathbf{p} + p_0^*\mathbf{1}$, then we have:

$$e^{i\theta^*} = \frac{(\mathbf{p}')^H\mathbf{q}}{|\mathbf{q}^H\mathbf{p}|} \quad (9)$$

If the lattice parameter is exempted from optimization, then $a^* = 1$. Otherwise:

$$a^* = \frac{|\mathbf{q}^H\mathbf{p}'|}{|\mathbf{q}|^2} \quad (10)$$

## Data availability

The data that support the findings of this study are available from the corresponding author.

## Code availability

Computation code is available from GitHub under https://github.com/johnaparker/miepy.

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

## Acknowledgements

The authors thank the Vannevar Bush Faculty Fellowship program sponsored by the Basic Research Office of the Assistant Secretary of Defense for Research and Engineering; Office of Naval Research (N00014-16-1-2502). We thank the Department of Chemistry for a Gustavus Swift Fellowship for one of us (CWP). We also acknowledge the University of Chicago Research Computing Center for providing the computational resources needed for this work.

## Author contributions

S.C. and N.F.S. wrote the paper. S.C. and C.W.P. performed the simulations. A.L.F. suggested the idea of HLDA. S.C. performed HLDA. C.W.P. performed the experiments and experimental data analysis. J.A.P. coded the simulation program. C.W.P., J.A.P., S.A. R., and A.L.F. critically revised the manuscript.

## Competing interests

The authors declare no competing interests.
