## [Peer Review File · Nature Communications]

REVIEWER COMMENTS

Reviewer #1 (Remarks to the Author):

This paper reports a very interesting and clearly meticulous study of the internal dynamics in 'optical matter'. The latter term is not widely familiar, but its usage is correct; it relates to multi-particle assemblies held together by forces that are most often referred to as 'optical binding'. The results are impressive and the analysis is clearly presented, with excellent figures and exceptionally extensive and clear captions. The conclusions are well founded and the literature citations are well chosen. In particular, the analysis and assumptions concerning compensation for rotational degrees of freedom are well argued.

Somewhat irritatingly, the online refereeing system gave me no access to the supplied video.

My main query, which I am sure the authors can take the opportunity to address and clarify in their paper, concerns the degree of appropriateness in their analogy to normal vibrational modes of molecules. It is indeed well known that low frequency modes such as antisymmetric stretches are highly relevant to many reaction intermediates, the classic three-body example being isotope exchange in diatomic hydrogen. But the key feature of molecular normal modes is a definitive oscillation frequency, and corresponding period. My impression is that in the overdamped positional fluctuations that lead to a kind of 2D isomerisation in this study, there would be no such regular period, is this right? Whether from experiment or simulation, taking large numbers of snapshot frames does not enable that distinction to be identified. So in my view the term 'normal mode' ought to be explained as not quite the same as is happening here.

I recommend publication subject to this clarification, and attention to the lesser points detailed below.

Although it should be obvious, the coordinates r in equation (1) are not defined.

On p.6 the authors write: 'the transitions are rare and detailed understanding requires higher sampling rates than can be readily obtained even in relatively high speed (100's fps) video measurements.' This seems to me an unconvincing reason to pursue the simulation route; it seems to be an unnecessary excuse based on a lack of adequate kit.

On p.7, it is stated that there is an 'inter-particle spacing of one optical wavelength', but is this quite accurate? Often in optical matter the best fit spacing is routinely a little different from one wavelength.

Reference is made to 'the two most stable configurations for six particles under a circularly polarized laser beam'. This needs to be clarified; I assume the term 'stable' signifies longest-lived (which is fine) but that should be distinguished from 'lowest energy', unless the latter is known or is being asserted.

On p.10 and in the caption to fig. 3, please define 'i.i.d. Gaussian'.

In fig. 3 the 20 mW results give the closest fit to theory, why were experiments not performed at this level? I imagine lower stability in the sense described above; this could usefully be explained.

Finally, another question: in fig. 6c, do these results mean that the transition mode is a linear combination of the shown modes, weighted by the exhibited displacements. with PC3 having much the largest magnitude coefficient? If this is so, it would be useful to explicitly say so; if not, please explain!

Reviewer #2 (Remarks to the Author):

I have read this manuscript "Data-Driven Reaction Coordinate Discovery in Overdamped and non-Conservative Systems: Application to Optical Matter Structural Isomerization" with much interest.

The authors provide a normal mode analysis study of colloidal clusters with interactions mimicking optical binding used in colloidal self-assembly. They focus on the isomerization transition from triangular to chevron shape of the colloidal "molecule".

Although the work is interesting, I have some doubts about the strong novelty claims put forward by the authors. I would like the authors to address my queries below before making a final decision about suitability of this paper for Nature Communications.

- The authors throughout the paper make statements such as "The primary goal of this paper is to develop an analog of normal mode analysis for non-conservative, overdamped systems..." or "There is no formal mechanical definition of normal modes for overdamped and non conservative systems."

I don't agree with the latter statement, and therefore I have some doubts about the novelty statement of the authors about the "primary goal" of their paper.

As a matter of fact, the normal modes of non-conservative systems with Langevin (damped or overdamped) dynamics have been studied numerically, in the context of mechanics of glassy systems. See for example: *Soft Matter*, 14, 8475-8482 (2018), *Macromolecules* 51, 4, 1559–1572 (2018) and *Phys. Rev. B* 102, 024108 (2020). In all these studies, the vibrational eigenmodes of particles undergoing Langevin dynamics in non-conservative (in fact, glassy and nonequilibrium) systems have been calculated.

These works should be mentioned and the authors should discuss the novelty of their approach compared to those works.

- I am quite surprised that the authors do not find negative eigenvalues in their analysis of what is prominently a relaxation process from a less stable to a more stable structure. These relaxational processes are associated with negative eigenvalues of the Hessian (i.e. purely imaginary eigenfrequencies), also known as instantaneous normal modes (INMs) in the chemical physics literature. In the above mentioned papers, the INMs of the non-conservative Langevin systems were carefully analyzed.

I am surprised that such analysis has not been performed here.

Surely there must be some negative eigenvalue(s) that underlies the relaxation from triangular to chevron shape?

- Related to the above issue, I was wondering if the molecular transition studied by the authors can be described in terms of a Jahn-Teller effect.

- Can the authors please explain why the 4th mode, that has the same 'weight'/variance in PCA doesn't lead to any transition. Is it because of their chosen conditions or due to other reasons?

- I find it rather difficult to catch the physical meaning of Eq. 2. The authors should provide some sentences to help the reader to grasp the physical content of this equation.

Reviewer #3 (Remarks to the Author):

The manuscript proposes a method based on the PCA to derive reaction coordinates on systems with overdamped and non-conservative interactions. The method could be used to replace the analysis of the Hessian matrix that is conventionally used in Hessian systems. While the paper could be of potential interest, I find it demanding on several respects and I am thus not convinced it can be published in the present form.

I have the following remarks:

- the authors claim that no method alternative to the Hessian is available for overdamped and non-conservative interactions. This is not true. Consider for instance the J-matrix approach proposed by I. Procaccia and co-workers in the framework of granular media (see *Phys. Rev. Lett.* 123, 098003, 2019 and). The author should compare their method with the J-matrix approach.

- No comparison is made with a control case to assess the validity of the PCA in replacing the

Hessian matrix. The authors should consider a simple Hamiltonian example and compare the reaction coordinates obtained with the Hessian matrix with the those obtained by PCA.

- An interesting issue of the Hessian analysis that is not discussed here is the effect of external driving force that push the system through a local instability. In that case some eigenvalue go to zero, showing an instability. It would be interesting to see if the PCA method can reveal this loss of stability.

- It is disappointing that the method is not used to analyze experimental images. The authors should at least try and discuss the limitations of the approach. How many frames would be needed? With what resolution? What can we get otherwise?

-The manuscript is very difficult to read and the authors should make an effort to streamline the discussion. Example: the discussion of HLDA is very hard to follow since the authors assume that the reader is knowledgeable with the technique. This might not necessarily be the case for a general reader of Nature Comm.

We thank the reviewers for their thoughtful comments. As a result of responding to their comments, we believe that the manuscript has been substantially improved.

Reviewer #1 (Remarks to the Author):

This paper reports a very interesting and clearly meticulous study of the internal dynamics in ‘optical matter’. The latter term is not widely familiar, but its usage is correct; it relates to multi-particle assemblies held together by forces that are most often referred to as ‘optical binding’. The results are impressive and the analysis is clearly presented, with excellent figures and exceptionally extensive and clear captions. The conclusions are well founded and the literature citations are well chosen. In particular, the analysis and assumptions concerning compensation for rotational degrees of freedom are well argued.

Author Reply: We thank the reviewer for their careful reading of our manuscript and warm reception of our work.

Somewhat irritatingly, the online refereeing system gave me no access to the supplied video.

Author Reply: We regret that the reviewer was unable to access the movie and hope that they can access this in our resubmission.

My main query, which I am sure the authors can take the opportunity to address and clarify in their paper, concerns the degree of appropriateness in their analogy to normal vibrational modes of molecules. It is indeed well known that low frequency modes such as antisymmetric stretches are highly relevant to many reaction intermediates, the classic three-body example being isotope exchange in diatomic hydrogen. But the key feature of molecular normal modes is a definitive oscillation frequency, and corresponding period. My impression is that in the overdamped positional fluctuations that lead to a kind of 2D isomerization in this study, there would be no such regular period, is this right? Whether from experiment or simulation, taking large numbers of snapshot frames does not enable that distinction to be identified. So in my view the term ‘normal mode’ ought to be explained as not quite the same as is happening here.

Author Reply: The reviewer is correct in that since the system is overdamped, there is no regular period in the particle motions so that what we have done is intrinsically different from conventional normal mode analysis. We had only intended that this be an analogy since the systems where normal modes are appropriate to use (e.g., bound molecules) are fundamentally different than optical matter. Still, the OM system can obtain and maintain defined structures with fluctuations (not vibrations) in harmonic-like wells due to optical binding. In order to make the relationship between the concept of normal modes and the collective modes we focus on for OM systems clear in this work, the changes below are now made to the manuscript.

First, a math derivation of the relationship between the normal modes and the statistical collective modes is added to the Supporting Information and mentioned in the paragraph just before Fig. 1 as “(a detailed derivation provided in **Supporting Information**)”.

Second, we have revised the Abstract: “Optical matter (OM) systems consist of (nano-)particle constituents in solution that can self-organize into ordered arrays that are bound by electrodynamic interactions. They also manifest non-conservative forces, and the motions of the nano-particles are overdamped; i.e., they exhibit diffusive trajectories. We propose a data-driven approach based on principal components analysis (PCA) to determine the collective modes of non-conservative overdamped systems, such as OM structures, and harmonic linear discriminant analysis (HLDA) of time trajectories to estimate the reaction coordinate for structural transitions. We demonstrate the approach via electrodynamic-Langevin dynamics simulations six electrodynamically-bound nanoparticles coupled to an incident laser beam. The reaction coordinate we discover is in excellent accord with a rigorous committor analysis, and the identified mechanism is in good agreement with the experimental observations. The PCA-HLDA

approach to data-driven discovery of reaction coordinates aid in understanding and eventually controlling non-conservative and overdamped systems including optical and active matter systems.”

Third, the following sentences have been added to the paragraph below Fig. 1: “Normal modes are orthogonal collective motions of particles that carry independent contributions to the system energy. The conventional definition of normal modes is valid only for harmonic particle-particle interactions.”

Fourth, the second paragraph of “Discussion and Conclusions” has been removed as we agree with the Reviewer's sentiment that this text pushed the analogy too far.

I recommend publication subject to this clarification, and attention to the lesser points detailed below.

Although it should be obvious, the coordinates \mathbf{r} in equation (1) are not defined.

Author Reply: We have now explicitly noted after Equation (1) that \mathbf{r} refers to the Cartesian coordinates of the molecular configuration.

On p.6 the authors write: ‘the transitions are rare and detailed understanding requires higher sampling rates than can be readily obtained even in relatively high speed (100’s fps) video measurements.’ This seems to me an unconvincing reason to pursue the simulation route; it seems to be an unnecessary excuse based on a lack of adequate kit.

Author Reply: It is true that experimental data is not used directly in the PCA-HLDA method because there are several technical challenges associated with its application to experimental measurements. The first is the magnitude of the data required. We show in **Fig. 5(b)** that the eigenvalues of the PCs take around 10^6 time steps (i.e., frames) to obtain converged results for this 6-nanoparticle OM system. Although it is technically possible to acquire a sufficient volume of experimental images with the correct number of particles, in practice it is exceedingly difficult to accumulate an adequate number of samples of rare transitions for a statistically robust analysis. Second, treatment of the experimental data poses an additional challenge due to low frame rates. Our current implementation of PCA-HLDA was developed for simulated trajectories with arbitrarily high frame rates (limited only by our integration time step) such that the identify of each particle between successive frames is identifiable (i.e., the frame rate is sufficiently high that the trajectory of each particle can be individually tracked). Under the lower frame rate currently available to us with our existing experimental setup, this is not possible since particles can undergo large rearrangements between frames, and we cannot unambiguously track the trajectory of each individual particle. As such, correctly assigning individual particles to specific lattice sites in experimental recordings is not currently possible. We agree with the reviewer that it is our aspiration to be able to rigorously study structural transitions in experimental trajectories in future research and have plans to obtain the necessary high-frame rate equipment. One of the primary outcomes of the present simulation-based work is to demonstrate the viability of the PCA-HLDA approach and lend confidence to its success in future applications to high-frame rate experimental data.

In order to clarify the challenges and aspirations, the following changes have been made to the manuscript:

First, the second paragraph after Fig. 1 has been changed to: “We demonstrate this PCA-HLDA approach described above in an application to trajectories of the triangle-to-chevron transition, like the measured result of Fig. 1(e-g), using combined electrodynamics-Langevin dynamics simulations of a 6-particle OM system. We note that the local fluctuation trajectory required for our PCA-HLDA approach must be adequately long and with sufficiently fine time steps. Although this analysis could in principle be done by particle tracking analysis of experimental data, the transitions are rare and must be sampled at rates that are higher than can be readily obtained even in relatively high speed (100’s fps) video measurements. We determine the contributions of each PCA collective mode to the transition, employ HLDA to formulate a reaction coordinate from these modes, validate the reaction coordinate using committor probability

analysis, and use our results to define the transition state ensemble and reaction mechanism. This PCA-HLDA approach is analogous to those used to describe molecular reaction dynamics, but it is herein applied to an overdamped and non-conservative OM system.”

Second, “by developing a method to consistently assign experimentally obtained particle positions to specific lattice sites for the trajectories obtained from experiments that involve rather frequent particle rearrangements” has been added at the end of the third paragraph of “Discussion and Conclusions”.

On p.7, it is stated that there is an ‘inter-particle spacing of one optical wavelength’, but is this quite accurate? Often in optical matter the best fit spacing is routinely a little different from one wavelength.

Author Reply: The reviewer is correct that there is a ~5% deviation between the optical wavelength and the inter-particle spacing for our 6-particle OM system. To be rigorous, the phrase “approximately” is now added before “one optical wavelength”. This is consistent with observations by others (Han, F.; Yan, Z. Phase Transition and Self-Stabilization of Light-Mediated Metal Nanoparticle Assemblies. *ACS Nano* **2020**, *14*, 6616-6625). This citation is also added to the manuscript.

Reference is made to ‘the two most stable configurations for six particles under a circularly polarized laser beam’. This needs to be clarified; I assume the term ‘stable’ signifies longest-lived (which is fine) but that should be distinguished from ‘lowest energy’, unless the latter is known or is being asserted.

Author Reply: Thank you for pointing this out. The term ‘most stable’ needs clarification. By ‘most stable’ we intend not necessarily the longest-lived but rather the most commonly observed or most probable (configuration). Below is the clarification that is now added right after the sentence that first contains ‘stable’ at the end of the second paragraph of “Results and Discussion”:

“Because energy is not well defined in the OM system, by saying a configuration is more stable we mean that it is more probable (more commonly observed in trajectories obtained from both simulation and experiment).”

On p.10 and in the caption to fig. 3, please define ‘i.i.d. Gaussian’.

Author Reply: The term ‘i.i.d. Gaussian’ is now defined as independent identical Gaussian distribution when it first appears in the 5th paragraph of ‘Results and Discussion’.

In fig. 3 the 20 mW results give the closest fit to theory, why were experiments not performed at this level? I imagine lower stability in the sense described above; this could usefully be explained.

Author Reply: The χ^2 distribution curve, which the 20 mW results give the closest fit to, is the theoretical result for an artificial 6-particle system in which the particle positions are fluctuating locally and independently (with no correlated motion). This is stated in the sentence in the manuscript (in the 5th paragraph of ‘Results and Discussion’):

“If in an artificial system where the particle motions are uncorrelated and described by independent identical Gaussian distributions (i.i.d. Gaussian), then the CDF of squared translationally and rotationally-minimized deviations should follow a χ^2 distribution with $(12-4) = 8$ degrees of freedom (black curve).”

The low-power regime of uncorrelated particle motions does indeed give good agreement with an independent-particle theoretical treatment but is not a terribly interesting regime since in the absence of correlated multi-body motions there are no collective modes of OM system. The primary focus of the present work is under higher-power conditions that induce correlated motions and interesting collective modes but where no good theoretical treatments currently exist. We state this a little further down in the paragraph:

"Again, the fitting displacement distribution shows increasing deviations from the 8 degree of freedom χ^2 distribution as the intensity of the optical field increases. This further supports the result that the extent of correlated motion of the particles increases with the intensity of the optical trapping beam."

Finally, another question: in fig. 6c, do these results mean that the transition mode is a linear combination of the shown modes, weighted by the exhibited displacements. with PC3 having much the largest magnitude coefficient? If this is so, it would be useful to explicitly say so; if not, please explain!

Author Reply: The reviewer is correct that the transition mode is a linear combination of the PC modes in terms of a position deviation vector and the coefficient of PC3 is largest in magnitude. The following changes has been made to the paragraph before Fig. 6 to clarify this for the reader.

First, the following text has been added: "Since the PCs are a complete basis of the space of position deviation vectors, the position deviation vector of each configuration can be written as a linear combination of the PCs, in which the coefficients can be obtained by orthogonal projection:

$$\mathbf{r} - \mathbf{r}_0 = \sum [\mathbf{v}_i^T (\mathbf{r} - \mathbf{r}_0)] \mathbf{v}_i$$

where, the \mathbf{v}_i 's are the principal components obtained from PCA, \mathbf{r}_0 is the coordinate of the triangle configuration, and \mathbf{r} is the coordinate of a specific configuration in the trajectory."

Second, the sentence "PC 3 dominates in the linear combination coefficients when configurational deviations are projected onto the PCs" has been added.

Reviewer #2 (Remarks to the Author):

I have read this manuscript "Data-Driven Reaction Coordinate Discovery in Overdamped and non-Conservative Systems: Application to Optical Matter Structural Isomerization" with much interest.

The authors provide a normal mode analysis study of colloidal clusters with interactions mimicking optical binding used in colloidal self-assembly. They focus on the isomerization transition from triangular to chevron shape of the colloidal "molecule".

Although the work is interesting, I have some doubts about the strong novelty claims put forward by the authors. I would like the authors the address my queries below before making a final decision about suitability of this paper for Nature Communications.

Author Reply: We thank the reviewer for their time and effort in considering our work and glad that they find it to be of interest. We are pleased to have the opportunity to respond to the points raised below.

- The authors throughout the paper make statements such as "The primary goal of this paper is to develop an analog of normal mode analysis for non-conservative, overdamped systems..." or "There is no formal mechanical definition of normal modes for overdamped and non-conservative systems."

I don't agree with the latter statement, and therefore I have some doubts about the novelty statement of the authors about the "primary goal" of their paper.

As a matter of fact, the normal modes of non-conservative systems with Langevin (damped or overdamped) dynamics have been studied numerically, in the context of mechanics of glassy systems. See for example: Soft Matter, 14, 8475-8482 (2018), Macromolecules 51, 4, 1559-1572 (2018) and Phys. Rev. B 102, 024108 (2020). In all these studies, the vibrational eigenmodes of particles undergoing Langevin dynamics in non-conservative (in fact, glassy and nonequilibrium) systems have been calculated.

These works should be mentioned and the authors should discuss the novelty of their approach compared to those works.

Author Reply: The non-conservative nature of the OM system discussed in this work refers to the electrodynamic forces created and acting on the particles (see: Sukhov, S.; Dogariu, A. Non-conservative Optical Forces *Rep. Prog. Phys.* **2017**, *80*, 112001). In other words, potential (energy) is not well-defined for the OM system with or without friction. This causes OM systems to have two intrinsic difference from the systems discussed in the articles cited by the reviewer, where the concept of potential (energy) is used. First, since the potential is not well-defined in our system, the Hessian matrix can only be calculated by taking the first derivative of the external force field instead of taking the second derivative of the potential. Second, the Hessian matrix generated in our system is asymmetric so that its eigenvalues and the eigenvectors are complex, its eigenvectors are not orthogonal to each other, and its left eigenvectors are different from its right eigenvectors. In order to clarify these issues, motivate the need for the present approach, and contextualize it within the prior literature, the changes below are now made to the paragraph after Fig. 1 of the manuscript:

First, the text following has been added: “In other words, if the following Langevin equation is considered:

$$m \frac{d^2 \mathbf{r}}{dt^2} = \mathbf{F}_{ext}(\mathbf{r}, t) - \xi \frac{d\mathbf{r}}{dt} + \boldsymbol{\eta}$$

where \mathbf{r} is the position, m is the mass, \mathbf{F}_{ext} is the external force field, ξ is the friction coefficient, and $\boldsymbol{\eta}$ is the random force, only when $\xi = 0$ and $\frac{\partial F_{ext,x}}{\partial y} = \frac{\partial F_{ext,y}}{\partial x}$ can normal modes be well-defined. It should be emphasized that the non-conservative nature of the OM system refers to the non-conservative (external) electrodynamic force field \mathbf{F}_{ext} .”

Second, the following text has been added: “Zaccone, et al. have used instantaneous normal modes (INM) to analyze non-affine dynamics of amorphous materials such as glassy polymers [56-58]. These two types of systems can be overdamped but are conservative (ξ is large enough to neglect the acceleration term; $\frac{\partial F_{ext,x}}{\partial y} = \frac{\partial F_{ext,y}}{\partial x}$). See the Supplementary Information for further discussion.”

Third, the following sentence has been added: “In our approach we define collective modes in overdamped and non-conservative systems (ξ is large enough to neglect the acceleration term; $\frac{\partial F_{ext,x}}{\partial y} \neq \frac{\partial F_{ext,y}}{\partial x}$).”

- I am quite surprised that the authors do not find negative eigenvalues in their analysis of what is prominently a relaxation process from a less stable to a more stable structure. These relaxational processes are associated with negative eigenvalues of the Hessian (i.e. purely imaginary eigenfrequencies), also known as instantaneous normal modes (INMs) in the chemical physics literature. In the above mentioned papers, the INMs of the non-conservative Langevin systems were carefully analyzed. I am surprised that such analysis has not been performed here.

Surely there must be some negative eigenvalue(s) that underlies the relaxation from triangular to chevron shape?

Author Reply: As stated in the reply to the question immediately above, the Hessian in the OM system is asymmetric so that its eigenvalues and eigenvectors are complex, its eigenvectors are not orthogonal to each other, and its left and right eigenvectors are different. This makes the math of the INMs (orthogonal and

real) stated in the papers mentioned by the reviewer not applicable in the OM system. The positive nature of the eigenvalues can also be understood by assuming for a moment that this were a conservative system, in which case the chevron and triangle configurations would correspond to metastable states residing in local potential energy minima and separated by an energy barrier. The eigenanalysis is performed for the structures associated with the metastable states, which explains the eigenvalues are always positive. If performed at the barrier, we would expect to obtain negative eigenvalues.

- Related to the above issue, I was wondering if the molecular transition studied by the authors can be described in terms of a Jahn-Teller effect.

Author Reply: The definition of Jahn-Teller effect requires the system to be Hamiltonian. Since all external forces besides friction are non-conservative in our system, its potential energy is not well-defined. Therefore, there is no well-defined Hamiltonian or energy difference between configurations. Therefore, we believe that Jahn-Teller effect is not applicable here.

- Can the authors please explain why the 4th mode, that has the same 'weight'/variance in PCA doesn't lead to any transition. Is it because of their chosen conditions or due to other reasons?

Author Reply: To clarify this issue, the following text has been added to the paragraph immediately above Fig. 6:

Note that the variance shown in Fig. 4 and the projection magnitudes shown in Fig. 6(c) are descriptions of two different processes. Fig. 4 describes a trajectory that contains only triangle configuration and its local fluctuations with no pattern transition. Fig. 6(c) describes an ensemble of trajectories of a specific pattern transition from triangle to chevron. Therefore, the large variance of the 4th mode described in Fig. 4 (local fluctuation of triangle) has no direct relation to whether it leads to the transition considered in Fig. 6(c) (transition from triangle to chevron). In addition, Fig. 6(c) only describes one specific transition starting from triangle (to chevron), so it is certainly possible that the 4th mode may dominate transitions to other states.

- I find it rather difficult to catch the physical meaning of Eq. 2. The authors should provide some sentences to help the reader to grasp the physical content of this equation.

Author Reply: An interpretation of the HLDA coordinate in terms of the Rayleigh ratio has been added below Eq. 2 in the manuscript. The text we added is:

“This reaction coordinate can be considered as the projection of the coordinate in the collective mode basis onto the vector \mathbf{W}^* that maximizes the Rayleigh ratio:

$$\mathbf{W}^* = \operatorname{argmax}_{\mathbf{W}} \frac{\mathbf{W}^T \mathbf{S}_b \mathbf{W}}{\mathbf{W}^T \mathbf{S}_w \mathbf{W}} = (\boldsymbol{\Sigma}_A^{-1} + \boldsymbol{\Sigma}_B^{-1})(\boldsymbol{\mu}_A - \boldsymbol{\mu}_B)$$

where $\mathbf{S}_b = (\boldsymbol{\mu}_A - \boldsymbol{\mu}_B)(\boldsymbol{\mu}_A - \boldsymbol{\mu}_B)^T$ is the between class scatter matrix, and $\mathbf{S}_w = (\boldsymbol{\Sigma}_A^{-1} + \boldsymbol{\Sigma}_B^{-1})^{-1}$ is the within class scatter matrix [4]. The vector \mathbf{W}^* can therefore be interpreted as the direction along which the two classes are best separated [4].”

Reviewer #3 (Remarks to the Author):

The manuscript proposes a method based on the PCA to derive reaction coordinates on systems with overdamped and non-conservative interactions. The method could be used to replace the analysis of the Hessian matrix that is conventionally used in Hessian systems. While the paper could be of potential interest, I find it demanding on several respects and I am thus not convinced it can be published in the present form.

Author Reply: We thank the reviewer for taking the time to evaluate our manuscript and pleased that they find it to be of interest. We regret that they found our presentation to be demanding and are pleased to have the opportunity to respond to the points raised below.

I have the following remarks:

- the authors claim that no method alternative to the Hessian is available for overdamped and non-conservative interactions. This is not true. Consider for instance the J-matrix approach proposed by I. Procaccia and co-workers in the framework of granular media (see Phys. Rev. Lett. 123, 098003, 2019 and). The author should compare their method with the J-matrix approach.

Author Reply: The reviewer is correct that the J-matrix approach is another important method that can be compared to this work. The J-matrix method leads to discovery of oscillatory solutions in the paper mentioned by the reviewer, which indeed sounds promising in analyzing non-conservative systems, *but is limited to underdamped systems*. In the overdamped case relevant for optical matter systems, the Langevin equation is first order so that there is no oscillatory solution. Therefore, in the overdamped OM system, this J-matrix method is still applicable, but since the absence of the oscillatory solution, the application is only limited to the analysis of its eigenvectors and eigenvalues. Next, since the J-matrix is asymmetric, its eigenvectors are not orthogonal to each other and its left eigenvectors are different from its right eigenvectors, which induces coupling between the collective modes represented by its eigenvectors. On the other hand, the covariance matrix analyzed using PCA gives orthogonal eigenvectors that make an orthogonal basis much easier to manipulate.

The text above has also been added to Supplementary Information.

The comparison above between the J-matrix method and our PCA-HLDA method is now added to the paragraph after Fig. 1 with the paper mentioned by the reviewer cited in the manuscript:

First, the following text has been added: “In other words, if the following Langevin equation is considered:

$$m \frac{d^2 \mathbf{r}}{dt^2} = \mathbf{F}_{ext}(\mathbf{r}, t) - \xi \frac{d\mathbf{r}}{dt} + \boldsymbol{\eta}$$

where \mathbf{r} is the position, m is the mass, \mathbf{F}_{ext} is the external force field, ξ is the friction coefficient, and $\boldsymbol{\eta}$ is the random force.”

Second, the following two sentences have been added: “Chattoraj, et al. have analyzed the eigenvalues and eigenvectors of J-matrix, the first derivative matrix of the force field, and found oscillatory solution of motion that are particularly useful for studies of underdamped non-conservative systems (ξ is not large enough to neglect the acceleration term; $\frac{\partial F_{ext,x}}{\partial y} \neq \frac{\partial F_{ext,y}}{\partial x}$). In overdamped cases, however, there exists no oscillatory solution while the unorthogonal eigenvectors of the J-matrix lead to intrinsically coupled collective modes that are complicated to analyze. See the Supplementary Information for additional discussion.”

- No comparison is made with a control case to assess the validity of the PCA in replacing the Hessian matrix. The authors should consider a simple Hamiltonian example and compare the reaction coordinates obtained with the Hessian matrix with the those obtained by PCA

Author Reply: A mathematical derivation of the relationship between the normal modes and the statistical collective modes obtained by PCA for conservative systems is now added to the Supporting Information, which is mentioned in the paragraph above Fig. 1 in the manuscript. The comparison requested is appropriate only when the Hessian matrix defines normal modes under deterministic motion and harmonic potential

energy function. For non-conservative systems, only by analogy to the conservative case do we use the statistical PCA approach. This is the only option we have because our system does not have a defined potential energy due to the non-conservative nature of the electromagnetic coupling. It is dissipative both via that coupling and solvent friction, and the motion is diffusive. Nevertheless, we show that one can define, using the PCA analysis, independent orthogonal collective coordinates that are the eigenvectors of the particle displacement covariance matrix that is obtained from the full set of particle trajectories. Although the displacement covariance matrix is quadratic in the displacements along the particle trajectories, the input data reflect the real forces in the system and the eigenvectors and eigenvalues derived from a PCA are different from – and provide different information than is encoded in – the normal mode eigenvectors and eigenvalues. Full details of this comparison are provided within the Supplementary Information.

- An interesting issue of the Hessian analysis that is not discussed here is the effect of external driving force that push the system through a local instability. In that case some eigenvalues go to zero, showing an instability. It would be interesting to see if the PCA method can reveal this loss of stability.

Author Reply: The reviewer's idea is indeed interesting and perhaps is a promising direction for future work, but one that we regard to be beyond the scope of the current manuscript. The transitions in this current work are between metastable states in the system. In other words, all (the real parts of) the eigenvalues of the Hessian (J -) matrix are positive. It would happen that under some changes in the optical field a certain pattern might be turned from metastable to unstable. However, so far, neither our experiments or simulations have explored such phenomena. In addition, PCA, which can only be applied to local metastable states, fails when analyzing unstable states because local fluctuation is not attainable for an unstable state.

- It is disappointing that the method is not used to analyze experimental images.

Author Reply: The experimental data-directed counterpart of this work will be a part of future work because we are not able to collect suitably large and precise experimental data at this time due to limitations of detectors (i.e. sensitivity and frame rate). This future ambition is now emphasized at the end of the 4th paragraph of "Discussion and Conclusions" in the manuscript. Please also see our response to a similar point raised in Comment #5 by Reviewer #1.

-The authors should at least try and discuss the limitations of the approach. How many frames would be needed? With what resolution? What can we get otherwise?

Author Reply: Although the limitations of the approach were already discussed together with future improvements in the 4th and 5th paragraph of the "Discussion and Conclusions" in the manuscript, we thank the reviewer for the prompt as there is more that we can say.

The following text has been added to fourth paragraph of "Discussion and Conclusions": "There are several challenges associated with implementing the PCA-HLDA method. The first is the magnitude of the data required. We show in Fig. 5b that the eigenvalues of the PCs take around 10^6 frames to converge for this system (with a convergence resolution 4%). While it is technically possible to acquire this number of experimental frames with the correct number of particles, the treatment of the data poses an additional challenge. Our current implementation of PCA-HLDA was developed for short simulated trajectories where each particle is assigned to a specific lattice site. The experimental data will contain several rearrangements where lattice assignment will switch. Developing a method to consistently assign experimentally obtained particle positions to specific lattice sites is not trivial."

-The manuscript is very difficult to read and the authors should make an effort to streamline the discussion. Example: the discussion of HLDA is very hard to follow since the authors assume that the reader is knowledgeable with the technique. This might not necessarily be the case for a general reader of Nature Comm.

Author Reply: We apologize for the wording in the manuscript that is difficult to read and have made efforts in our responses to reviewer comments to simplify and improve our presentation for the reader. In regard to the particular issue of HLDA, we have added the following text below Eq. 2 in the manuscript to provide an interpretable explication of the approach for the unfamiliar reader:

“This reaction coordinate can be considered as the projection of the coordinate in the collective mode basis onto the vector \mathbf{W}^* that maximizes the Rayleigh ratio:

$$\mathbf{W}^* = \operatorname{argmax}_W \frac{\mathbf{W}^T \mathbf{S}_b \mathbf{W}}{\mathbf{W}^T \mathbf{S}_w \mathbf{W}} = (\boldsymbol{\Sigma}_A^{-1} + \boldsymbol{\Sigma}_B^{-1})(\boldsymbol{\mu}_A - \boldsymbol{\mu}_B)$$

where $\mathbf{S}_b = (\boldsymbol{\mu}_A - \boldsymbol{\mu}_B)(\boldsymbol{\mu}_A - \boldsymbol{\mu}_B)^T$ is the between class scatter matrix, and $\mathbf{S}_w = (\boldsymbol{\Sigma}_A^{-1} + \boldsymbol{\Sigma}_B^{-1})^{-1}$ is the within class scatter matrix [4]. The vector \mathbf{W}^* can therefore be interpreted as the direction along which the two classes are best separated [4].”

REVIEWERS' COMMENTS

Reviewer #1 (Remarks to the Author):

It is rewarding to see the meticulous attention that the authors have given to each and every point raised by the three reviewers, each of whom had given substantive feedback. In my view every issue has now been addressed, with the authors making a few corrections, clarifications and deletions, and importantly by adding explanatory text at all the necessary junctures in the paper. This is a fine piece of work, made much clearer through this revision, and it is now eminently well suited to publication.

Reviewer #2 (Remarks to the Author):

I am quite satisfied with the authors' rebuttal. With this extensive revision the manuscript is probably publishable in Nature Commun. The authors should note new developments in the mathematical modelling of eigenmodes of systems with overdamped dynamics, e.g. PNAS 118 (5) e2022303118 (2021).

The authors should also carefully check the references, which contain several either misspelled or missing names e.g. 37, 38, 55 and possibly more.

Reviewer #3 (Remarks to the Author):

The authors have answered in details to all the concerns I raised. I think the paper is improved and could be accepted for publication.

We thank the reviewers for their thoughtful comments. As a result of responding to their comments, we believe that the manuscript has been substantially improved.

Reviewer #1 (Remarks to the Author):

It is rewarding to see the meticulous attention that the authors have given to each and every point raised by the three reviewers, each of whom had given substantive feedback. In my view every issue has now been addressed, with the authors making a few corrections, clarifications and deletions, and importantly by adding explanatory text at all the necessary junctures in the paper. This is a fine piece of work, made much clearer through this revision, and it is now eminently well suited to publication.

Author Reply: We thank the reviewer for their careful reading of our manuscript and warm reception of our work.

Reviewer #2 (Remarks to the Author):

I am quite satisfied with the authors' rebuttal. With this extensive revision the manuscript is probably publishable in Nature Commun.

Author Reply: We thank the reviewer for their time and effort in considering our work and glad that they find it to be of interest. We are pleased to have the opportunity to respond to the points raised below.

The authors should note new developments in the mathematical modelling of eigenmodes of systems with overdamped dynamics, e.g. PNAS 118 (5) e2022303118 (2021).

Author Reply: The reviewer's comment makes us aware that there are new developments in mathematical modeling of eigenmodes. Therefore, we added the following to the paragraph after Eq. 2 with the corresponding citations:

Zaccone and co-workers have used instantaneous normal modes (INM) and the vibrational density of states in liquids to analyze none-affine dynamics of amorphous materials such as glassy polymers.

The authors should also carefully check the references, which contain several either misspelled or missing names e.g. 37, 38, 55 and possibly more.

Author Reply: We thank the reviewer for pointing out the typos in references. We have checked through the reference section and fixed all typos.

Reviewer #3 (Remarks to the Author):

The authors have answered in details to all the concerns I raised. I think the paper is improved and could be accepted for publication.

Author Reply: We thank the reviewer for taking the time to evaluate our manuscript and pleased that they find it to be of interest.